# Waveform Optimization Control of an Active Neutral Point Clamped Three-Level Power Converter System

**Jinghua Zhou and Jin Li \***

Beijing Laboratory of New Energy Storage Technology, Beijing 100144, China
* Correspondence: 2019413010106@mail.ncut.edu.cn

**Abstract:** Currently, the escalating integration of renewable energy sources is causing a steady weakening of grid strength. When grid strength is weak, interactions between inverters or those between inverters and grid line impedance can provoke widespread oscillations in the power system. Additionally, the diverse DC voltage application characteristics of power converter systems (PCS) may lead to over-modulation, generating narrow pulse issues that further impact control of the midpoint potential balance. Existing dead-time elimination methods are highly susceptible to current polarity judgments, rendering them ineffective in practical use. PCS, due to inherent dead-time effects, midpoint potential imbalances in three-level topologies, and narrow pulses, can elevate low-order harmonic content in the output voltage, ultimately distorting grid-connected currents. This is particularly susceptible to causing resonance in weak grids. To enhance the output voltage waveform of PCS, this article introduces a comprehensive compensation control strategy that combines dead-time elimination, midpoint potential balance, and narrow pulse suppression, all based on an active neutral point clamped (ANPC) three-level topology. This strategy gives precedence to dead-time elimination and calculates the upper and lower limits of the zero-sequence available for midpoint potential balance while fully compensating for narrow pulses. By prioritizing dead-time elimination, followed by narrow pulse suppression and finally midpoint potential balance, this method decouples the coupling between these three factors. The effectiveness of the proposed method is validated through semi-physical simulations.

**Keywords:** active neutral point clamped topology; power converter system; waveform optimization; deadtime elimination; narrow pulse; neutral point potential balance

## 1. Introduction

With the rapid development of the global economy, the continuous increase in human consumption of energy, the gradual depletion of fossil energy, and increasingly serious environmental pollution have prompted continuous innovation in the energy structure, and the proportion of renewable energy in total energy is increasing. In 2018, the International Renewable Energy Agency (IRENA) released the "Global Energy Transition: Roadmap to 2050", which pointed out that in order to achieve the global carbon emissions target by 2050, the proportion of global renewable energy generation needs to increase from 26% to 55% by 2050 [1]. With the continuous development of renewable energy generation technologies, represented by photovoltaics and wind power, the power system is exhibiting a new trend characterized by high permeability of renewable energy integration and a high proportion of power electronic converters. In order to adapt to renewable energy's characteristics of randomness, intermittency, and volatility, multiple distributed energy sources, energy storage systems, and controllable loads are integrated and receive joint regulation to form a microgrid operation, which can shorten the power transmission distance and improve system efficiency. In microgrids, inverter-based resources (IBR) account for a relatively high proportion, and the grid strength decreases significantly. Due to the high line impedance, the coupling effects between individual IBRs and between IBRs and transmission lines have

a more serious impact on the stability of the microgrid, even leading to microgrid instability. Due to the dynamic response of different IBRs over multiple time scales, the disturbance frequency band between devices in the microgrid is broader; in addition, nonlinear links such as dead zones and amplitude limitations that are commonly present in IBRs will also have a certain impact on system stability, further exacerbating the dynamic complexity of the microgrid [2].

Battery Energy Storage Systems (BESS) can smooth out the output of renewable energy sources, support grid voltage and frequency, and become an effective means to address the issues arising from the high permeability of renewable energy. As the core component of BESS, PCS serve as the interface between the energy storage medium and the grid as well as the load. In energy storage systems with DC voltages ranging from 750 to 1500 V and power ratings from 50 kW to 3.5 MW, PCS mostly adopt a three-level topology, which offers better harmonic characteristics, lower dv/dt, lower device voltage tolerance, and higher conversion efficiency compared to two-level converters. Depending on different power and voltage ratings, topologies such as NPC, T-NPC, and ANPC can be selected.

Pulse-Width Modulation (PWM) is widely applied in the drive controls of PCS for renewable energy generation, due to its advantages such as high efficiency and low output harmonics. However, switching devices like insulated gate bipolar transistors (IGBT) have inherent turn-on and turn-off times. To prevent improper switching of IGBT in practical applications, a dead time must be inserted into the drive signal. However, the dead-time effect can cause deviations in the fundamental frequency components of the output current, as well as positive-sequence currents of the $6k + 1$ order and negative-sequence currents of the $6k - 1$ order. These harmonic currents can lead to distortion in the output current.

To address the dead-time issue, existing research focuses on two directions: dead-time compensation [3] and dead-time elimination [4]. Dead-time compensation involves obtaining a compensation voltage through device parameter tables or experimental methods and applying it based on the polarity of the current. However, this method faces several challenges: ① insufficient accuracy in judging the polarity of the current can lead to failed compensation [5]; ② the actual drive pulse lags behind the ideal drive pulse by half the dead time, resulting in errors in the instantaneous voltage [6]; ③ dead-time compensation can increase the amplitude of the modulation wave, potentially causing issues such as narrow pulses and other nonlinear factors [7]. In contrast, dead-time elimination works by only modifying the drive signal of the active power switch while keeping the corresponding complementary power switch inactive. Since the active drive signal does not contain a dead time, it fundamentally avoids the dead-time effect. However, the main challenge with dead-time elimination is the high accuracy required in judging the polarity of the current. Incorrect polarity judgments can lead to more severe output voltage distortion compared to dead-time compensation. Therefore, it is crucial to select the appropriate dead-time handling method for the specific application scenario [8].

When the modulation index of the PCS is high, the addition of dead time can easily result in individual drive signals being shorter than the minimum turn-on or turn-off time, leading to narrow pulses. This incomplete switching of power devices increases switching losses, causes output waveform distortion, and may even lead to thermal accumulation and burnout of the power devices. Common narrow pulse suppression methods include direct narrow pulse elimination, zero-sequence voltage injection, and non-nearest three-vector SVPWM modulation. The principle of the direct narrow pulse elimination method is to exclude or extend the pulse width when the pulse width of the PWM signal is less than a set limit, avoiding the generation of narrow pulses. However, this may lead to over-adjustment of the voltage in one phase while the voltages in other phases remain unchanged, disrupting the balance of the three-phase voltages. The zero-sequence voltage injection method adjusts the PWM waveform by injecting a zero-sequence voltage into the three-phase voltages to eliminate or expand narrow pulses. While maintaining constant line voltages, this method addresses narrow pulse issues by modifying the phase voltages, but this may introduce narrow pulses in other phases and increase the harmonic content of

the output voltage [9]. Additionally, the non-nearest three-vector method can be employed to avoid narrow pulses by replacing the basic vectors with action times shorter than the minimum pulse width. However, this method comes at the cost of increased switching frequency and a reduced distribution area for narrow pulses [10].

Furthermore, imbalance in the DC neutral-point potential is a core issue in three-level converters, and can be categorized into DC imbalance and AC oscillation [11]. The primary causes of DC imbalance can be summarized as follows: ① inconsistency in capacitor parameters [12]; ② asymmetric loading [10]; and ③ inherent characteristics of the control strategy [13]. Common compensation methods for DC imbalance introduce a third-harmonic zero-sequence voltage into the modulation wave, resulting in a triple-frequency ripple in the DC voltage, known as AC oscillation. The magnitude of the AC oscillation ripple depends on the output current, power factor, and DC capacitance. An excessive ripple can lead to distortion in the AC output voltage and excessive voltage stress on power devices.

Disturbance voltages generated by the nonlinear factors of PCS can cause distortion in the output voltage of the inverter, leading to adverse effects such as phase current distortion, torque ripple, and degraded control performance [14]. Currently, the mainstream control method for PCS involves synchronizing with the power grid through a phase-locked loop (PLL) and transmitting energy to the grid in the form of a current source. Under a strong grid, grid-following control (GFL) can achieve precise power control and obtain high grid-connected power quality. However, in microgrids with a high proportion of power electronics, the system has low inertia and insufficient support capabilities. Significant fluctuations in line impedance can lead to voltage fluctuations or even oscillations at the grid-connection point. Grid-following control can easily trigger grid-connected current resonance or even instability [15]. Additionally, due to three-phase imbalance and sudden load changes, PCS controlled by GFL are unable to provide inertia and voltage support to the microgrid, resulting in noticeable power quality issues such as current/voltage harmonics, imbalance, and voltage fluctuations [3].

This article aims to optimize the control of a three-level PCS in a microgrid. The primary contribution of this paper is the proposal of a comprehensive compensation control method for nonlinear factors in the ANPC three-level topology. Building upon existing dead-time elimination techniques for NPC three-level topologies, this method optimizes dead-time elimination for ANPC topologies, exhibiting robust fault tolerance against errors in current polarity. Even under high modulation indices, it achieves a favorable grid-connected current waveform. Furthermore, the paper harmonizes the zero-sequence voltage required for balancing the neutral-point potential and suppressing narrow pulses, providing an optimized range for zero-sequence voltage injection.

In this article, Section 1 introduces the application background of renewable energy power generation, and summarizes the current status of research on three-level nonlinear factors, including dead-time effect, DC neutral-point potential imbalance, narrow pulse, and other related issues. Section 2 analyzes the formation mechanism of the nonlinear output voltage waveform in ANPC three-level topology. Section 3 optimizes the control of three influencing factors, elaborates on the coupling relationships among various influencing factors, and proposes a comprehensive compensation control method for the nonlinear factors of the ANPC three-level topology. Section 4 validates the effectiveness of the proposed control algorithm through hardware-in-the-loop simulations, demonstrating its ability to effectively reduce the low-order harmonic content of the output voltage and grid-connected current under various fault conditions. Section 5 summarizes and provides an outlook for optimization methods for nonlinear factors.

## 2. Analysis of Nonlinear Factors of PCS

The topology of an ANPC three-level PCS is shown in Figure 1. Unlike the carrier modulation strategy of a two-level PCS, the carrier modulation of the ANPC topology mainly relies on carrier stacking algorithms. Narrow pulses may occur near the zero point

and peak of the modulation wave, causing the switching tubes to fail to turn on or off properly. Additionally, the introduction of dead time can exacerbate narrow pulses or even lead to pulse shaping, further causing nonlinear distortion in the output voltage and current. When a dead time is introduced, theoretically, the two complementary pulses are no longer strictly complementary, and there is a situation where both complementary power switches are at 0 during the dead time. At this point, the output voltage is related to the polarity of the current. When the current is greater than 0, the output voltage is $-U_{dc}/2$; when the current is less than 0, the output voltage is $-U_{dc}/2$.

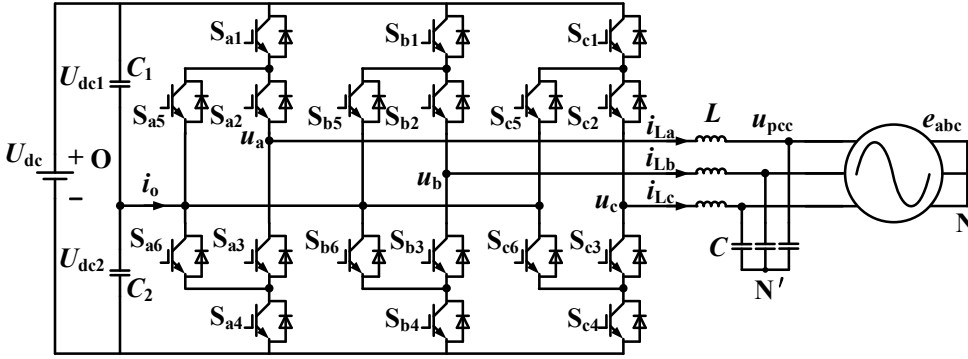

**Figure 1.** ANPC topology of PCS.

The average error voltage $\Delta V_d$ can be expressed as follows:

$$\Delta V_d = \frac{T^* - T_{on}}{T_s} = \begin{cases} (t_{on} - t_{off} - t_d)/T_s & (i > 0) \\ (t_{on} - t_{off} + t_d)/T_s & (i < 0) \end{cases} \tag{1}$$

In Equation (1), $T_s$ denotes the switching period, $T^*$ represents the theoretical turn-on time within $T_s$, while $T_{on}$ signifies the actual turn-on time. The symbol $i$ stands for the grid-connected current, with its reference direction aligning with that of $i_{La}$ depicted in Figure 1. Any deviation between $T^*$ and $T_{on}$ gives rise to an output voltage error, denoted as $\Delta V_d$. This $\Delta V_d$ is intricately linked to the power device's turn-on time $t_{on}$, turn-off time $t_{off}$, and dead time $t_d$. Furthermore, the polarity of $\Delta V_d$ is contingent on the polarity of $i$.

The average error voltage $\Delta V_d$ leads to an error current, which can be expressed as:

$$\begin{cases} \Delta \bar{i}_a = \frac{4(t_{on} - t_{off} - t_d)u_{dc}}{\pi T_s} \sum_{n=1,3,5,\dots} \frac{1}{n^2 \omega L_c} \sin(n\omega t - \frac{\pi}{2}) \\ \Delta \bar{i}_b = \frac{4(t_{on} - t_{off} - t_d)u_{dc}}{\pi T_s} \sum_{n=1,3,5,\dots} \frac{1}{n^2 \omega L_c} \sin[n(\omega t - \frac{2\pi}{3}) - \frac{\pi}{2}] \\ \Delta \bar{i}_c = \frac{4(t_{on} - t_{off} - t_d)u_{dc}}{\pi T_s} \sum_{n=1,3,5,\dots} \frac{1}{n^2 \omega L_c} \sin[n(\omega t + \frac{2\pi}{3}) - \frac{\pi}{2}] \end{cases} \tag{2}$$

As shown in Equation (2), the dead-time effect results in deviations in the fundamental frequency component of the output current, as well as the presence of $6k + 1$th-order positive-sequence currents and $6k - 1$th-order negative-sequence currents. These harmonic currents lead to distortion in the output current.

For three-level PCS, nonlinear factors such as dead time and DC neutral-point potential imbalance can influence the output waveform, introducing low-order harmonics. In power systems with high penetration of intermittent and renewable energy sources (IBR), when the frequency of these low-order harmonics coincides with the quasi-resonant frequency of the weak grid, the low-order harmonic currents can be amplified, significantly affecting the quality of the grid-connected power. The Norton equivalent circuit for a PCS grid-connected system is shown in Figure 2, where $i_{s\_\alpha\beta}(s)$ represents the equivalent current source of the inverter, while $Z_o(s)$ denotes its output impedance. The voltage at the grid-connected point is designated $u_{pcc\_\alpha\beta}(s)$, $u_{g\_\alpha\beta}(s)$ represents the grid voltage, and $Z_g(s)$ stands for the line impedance.

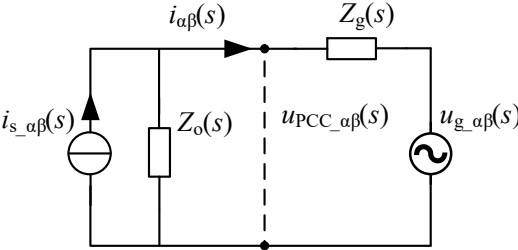

**Figure 2.** Norton equivalent circuit for PCS grid-connected systems.

The expression of the grid-connected current of the PCS can be obtained from Figure 2 as follows:

$$i_{\alpha\beta}(s) = [i_{s\_\alpha\beta}(s) - \frac{u_{g\_\alpha\beta}(s)}{Z_o(s)}]\frac{1}{1 + Z_g(s)/Z_o(s)} = A(s)B(s) \tag{3}$$

According to the impedance criterion, the $B(s)$ in Equation (3) can be regarded as a closed-loop transfer function with a forward channel gain of 1. Therefore, $Z_g(s)/Z_o(s)$ should satisfy the Nyquist stability criterion. As the dead time increases, the system's phase margin decreases, and the output impedance of the PCS approaches the grid line impedance towards $180°$. With the increase in line impedance, low-order resonance may occur in the grid-connected current, leading to a decrease in the quality of the grid-connected power.

From the above analysis, it can be seen that the nonlinear factors of the inverter mainly include dead-time effects, narrow pulse effects, and DC neutral point imbalance. These factors are coupled and jointly cause distortion in the output waveform of the inverter.

### 3. Principle of Dead Zone Elimination Method and Optimal Control of ANPC Three-Level PCS

*3.1. ANPC Three-Level PCS Dead Zone Elimination Method*

Figure 3 depicts the circulating current process in phase A of the ANPC PCS. In this figure, $u_a$ exhibits three distinct levels, corresponding to the P state, O state, and N state, when the AC output point is linked to the P point, O point, and N point of the DC bus capacitors, respectively. Depending on the positive and negative values of the modulation voltage $u_a{}^*$ and the current $i_a$, Figure 3 can be categorized into four quadrants within a single power frequency cycle. Specifically, Figure 3a depicts $u_a{}^* > 0$ and $i_a > 0$, Figure 3b shows $u_a{}^* < 0$ and $i_a > 0$, Figure 3c illustrates $u_a{}^* < 0$ and $i_a < 0$, and Figure 3d represents $u_a{}^* > 0$ and $i_a < 0$. Within each quadrant, two distinct stages are observed: the current charging stage and the current freewheeling stage. During the charging stage, the amplitude of the current rises, effectively charging the AC inductor ($L$ in Figure 1). Conversely, during the freewheeling stage, the current amplitude decreases as the AC inductor discharges its stored energy. $S_{a1}$~$S_{a6}$ represent the driving pulses that correspond to the IGBTs ($Q_{a1}$~$Q_{a6}$), with '0' indicating the low state and '1' indicating the high state of the driving pulses. Given that there are two alternative paths in the O state, there exist multiple modulation strategies that can achieve normal modulation. Here, we present the commonly used PWM-1 modulation method. To prevent short-circuiting in the converter, it is crucial to ensure that $S_{a1}$ and $S_{a5}$, $S_{a2}$ and $S_{a3}$, or $S_{a4}$ and $S_{a6}$ are not simultaneously turned on. Consider Figure 3a as an example. When $u_a{}^*$ is greater than or equal to 0 and $i_a$ is positive, the P state corresponds to the current boosting stage. In this stage, $S_{a1}$ and $S_{a2}$ are set to '1', $S_{a3}$, $S_{a4}$, and $S_{a5}$ are set to '0', and $S_{a6}$ is set to '1'. This arrangement ensures that IGBTs $S_{a3}$ and $S_{a4}$ share the same voltage ($U_{dc}/2$). Moving to the freewheeling stage depicted in Figure 3b, $S_{a1}$ flips to '0', and the current freewheels through the freewheeling diode of $S_{a5}$ to $S_{a2}$, and then through the freewheeling diode of $S_{a6}$ to $S_{a2}$. Notably, $S_{a5}$ exhibits pulse redundancy, meaning its switching state does not impact the commutation path. Similar observations can be made in other intervals, further confirming the existence of pulse redundancy during the current freewheeling stage.

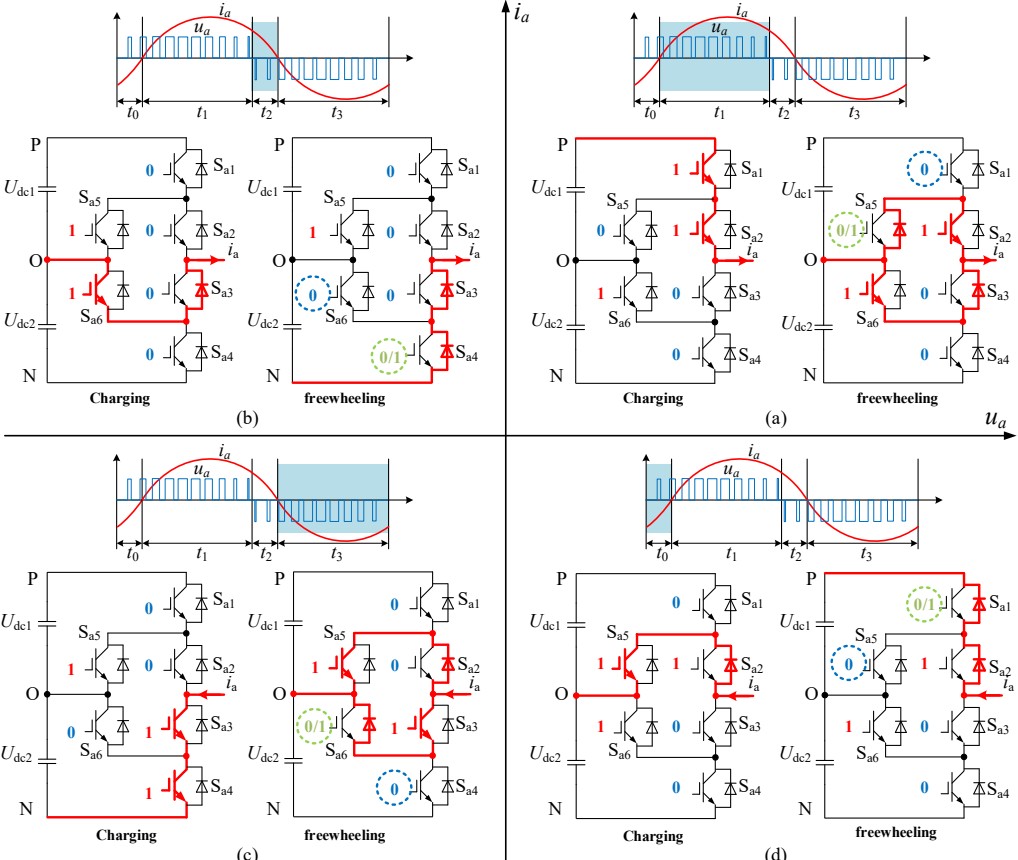

**Figure 3.** Current commutation diagram of ANPC three-level topology. (**a**) $i_a > 0$ and $u_a$ transitions between P and O states. (**b**) $i_a > 0$ and $u_a$ transitions between O and N states. (**c**) $i_a < 0$ and $u_a$ transitions between O and N states. (**d**) $i_a < 0$ and $u_a$ transitions between P and O states.

To avoid the dead-time effect, redundant states in the drive pulses can be eliminated based on the polarity of the original modulation wave $u_a{}^*$ and current $i_a$. This results in the PWM signals for $S_{a1}$ to $S_{a6}$ shown in Figure 4a. The fundamental mechanism for generating the drive pulses is similar to traditional carrier-based PWM. Drive pulses are generated by comparing $u_a{}^*$ with two in-phase stacked carriers, *Cr*+ and *Cr*−. Redundant drive pulses are alternately disabled based on the polarity of the reference current $i_a$ and modulation wave $u_a{}^*$. Specifically, $S_{a1}$ is disabled when $u_a{}^* > 0$ and $i_a < 0$, $S_{a5}$ is disabled when $u_a{}^* > 0$ and $i_a > 0$, $S_{a4}$ is disabled when $u_a{}^* < 0$ and $i_a > 0$, and $S_{a6}$ is disabled when $u_a{}^* < 0$ and $i_a < 0$.

Eliminating redundant states in the modulation depicted in the above figure brings significant advantages. Notably, $S_{a1}$ and $S_{a5}$, $S_{a2}$ and $S_{a3}$, and $S_{a4}$ and $S_{a6}$ will not simultaneously equal '1', essentially preventing the occurrence of current shoot-through. Consequently, the dead time between traditional complementary drive pulses can be eliminated. Moreover, the absence of dead-time effects reduces low-frequency output voltage distortion, while also decreasing the number of switching events, leading to lower switching losses. The PWM shown in Figure 4a exhibits a dead-time-free characteristic, referred to as dead-time elimination. However, traditional dead-time elimination methods heavily rely on accurate current polarity detection. As is evident in Figure 4b, when there is a delay in polarity judgment, polarity errors occur within the regions where the current is greater than 0, resulting in extra forbidden and enabled intervals for $S_{a1}$ and $S_{a3}$, respectively.

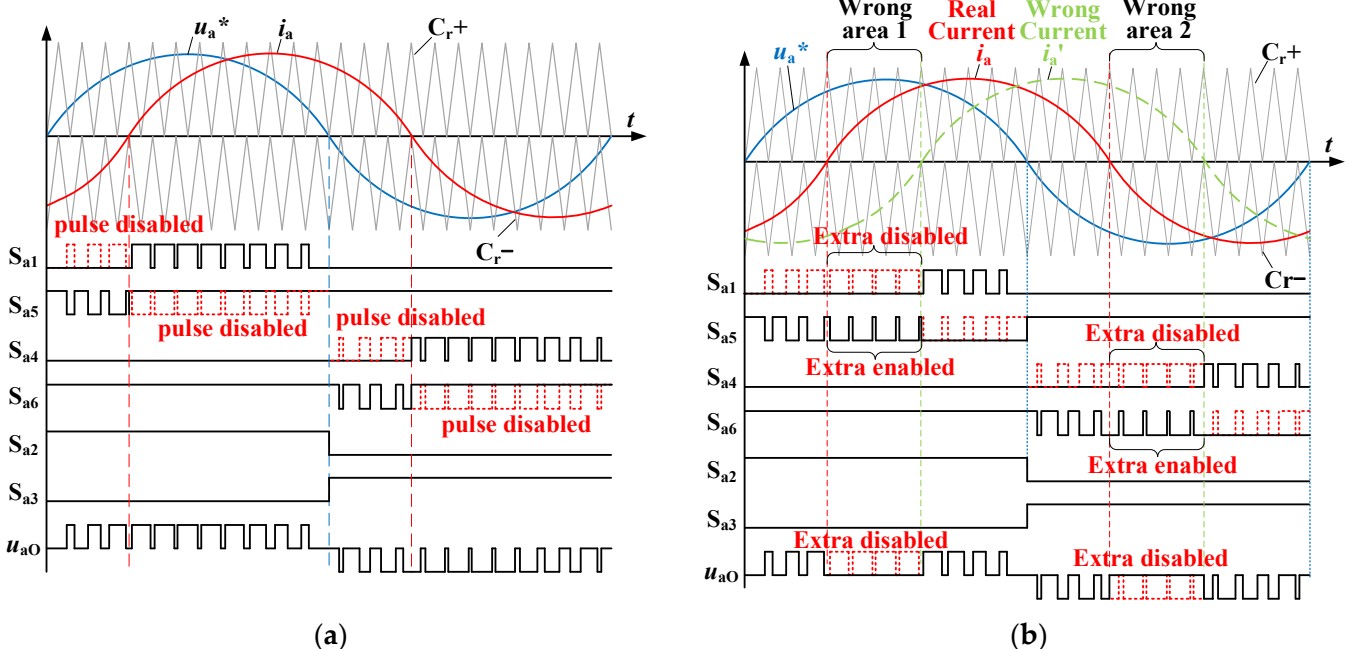

**Figure 4.** Principle of eliminating the dead zone in ANPC topology. (**a**) The basic principle of the dead-zone elimination method. (**b**) The effect of dead-band elimination on the output voltage $u_{ao}$ when there is a delay in current sampling $i_a'$.

Similarly, in regions where the current is less than 0, $S_{a4}$ and $S_{a6}$ experience similar polarity errors. These polarity errors lead to voltage gaps in the output voltage, causing voltage distortion. Additionally, due to the current pumping and freewheeling processes within a switching cycle, current ripple exists. This ripple can cause unexpected polarity jumps at the zero-crossing points, resulting in abnormal dead-time elimination judgments at these points.

### 3.2. ANPC Three-Level PCS Dead Zone Elimination Optimization Method

To retain the dead-time elimination feature while reducing dependency on current polarity, it is necessary to maintain the redundant states during the freewheeling phase of the drive pulse signals. In the event of a misjudgment of current polarity, the presence of these redundant states can effectively minimize errors in the output voltage. Building upon the work in reference [8], this paper proposes a dual modulation wave PWM technique for the ANPC three-level PCS. This paper introduces an additional modulation wave $u_a^{**}$ to complement the original modulation wave $u_a^*$. The magnitude of $u_a^{**}$ is adjusted based on the current polarity. Both modulation waves act concurrently during the current freewheeling phase, creating overlapping periods between the additional drive pulses and the original drive pulses of the dead-time elimination PWM. This approach prevents shoot-through and eliminates dead-time effects. The proposed dual modulation wave PWM is illustrated in Figure 5.

In Figure 5, an additional modulation wave $u_a^{**}$ is introduced alongside the original modulation wave $u_a^*$. $u_a^{**}$ generates redundant drive pulses during the current freewheeling phase. When $i_a \geq 0$, the magnitude of $u_a^{**}$ is increased by $\Delta u^*/2$ relative to $u_a^*$. Conversely, when $i_a < 0$, the magnitude of $u_a^{**}$ is reduced by $\Delta u^*/2$ relative to $u_a^*$. Within the original redundant regions, carrier modulation is performed based on $u_a^{**}$, with $S_{a1}$–$S_{a2}$–$S_{a4}$ and $S_{a3}$–$S_{a5}$–$S_{a6}$ using the same modulation wave. The specific distribution is indicated by the different colors of $S_{a1}$~$S_{a6}$ in Figure 5. As can be seen in the enlarged area of Figure 5, the amplitude difference $\Delta u^*$ between $u_a^{**}$ and $u_a^*$ creates an overlapping low-level time $T_u$, between the high levels of $S_{a1}$ and $S_{a5}$. This prevents shoot-through failures, and the high-level duration of the output voltage remains the same as that theoretically calculated.

Therefore, it can be concluded that the dual modulation wave PWM belongs to the category of dead-time elimination, being essentially free from dead-time effects. The added drive pulses, occurring during the freewheeling phase, do not affect the original output voltage of the dead-time elimination PWM.

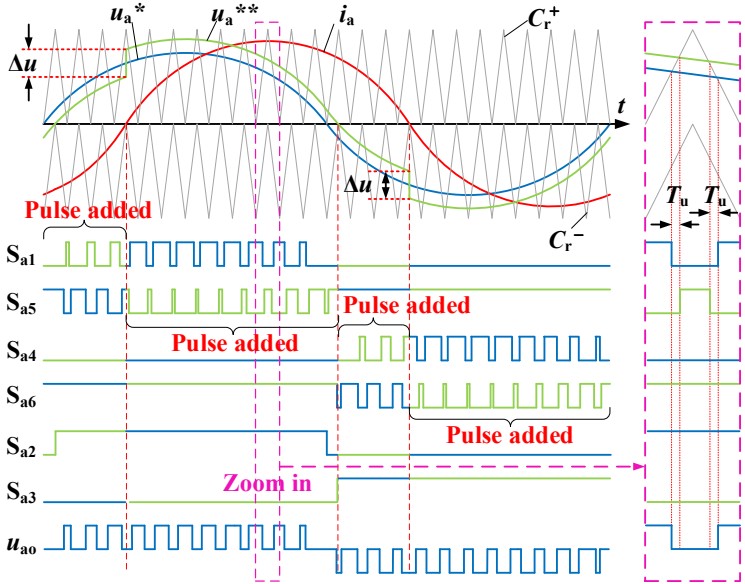

**Figure 5.** Principle of double modulation wave dead zone elimination for ANPC topology.

Based on the similarity triangle relationship shown in Figure 5, the increased amplitude of the modulation wave $\Delta u^*$ can be derived as follows:

$$\begin{cases} \Delta u = \dfrac{2T_u}{T_s}\dfrac{U_{dc}}{2} \\ T_u = t_d \end{cases} \tag{4}$$

In Equation (4), $T_s$ represents the switching period, and $U_{dc}$ is the DC bus voltage. The overlapping low-level time within a single switching period is $2T_u$. The $\Delta u^*$ is directly proportional to $T_u$ and inversely proportional to $T_s$. Unlike traditional dead-time, which can introduce dead-time effects, $T_u$ does not affect the output voltage. Therefore, $T_u$ can be set as the dead-time $t_d$.

In the ANPC dual modulation wave PWM, when the modulation index is relatively high, the generated amplitude may exceed the carrier, causing over-modulation. As can be seen in Figure 6a, over-modulation only occurs in the modulation wave $u_a^{**}$ used during the freewheeling phase, resulting in the disappearance of the redundant drive pulses added during the current freewheeling stage. However, the freewheeling phase does not affect the output voltage, so it does not cause output current distortion issues even when the modulation index is high. Furthermore, the proposed method exhibits better fault tolerance for current polarity judgments compared to conventional dead-time elimination methods. As illustrated in Figure 6b, even in the event of a current polarity error, the PWM pulse is enabled during the freewheeling phase. Although this pulse deviates from the theoretical pulse by an amplitude of $\Delta u^*$, this deviation only generates two voltage error pulses within a single switching period of the output voltage. This is a significant improvement compared to Figure 6b, where more noticeable voltage loss occurs in the output voltage during regions of incorrect current polarity. Therefore, in areas where current polarity misjudgments are more frequent, such as near the current zero-crossing points, the proposed method can significantly improve output voltage and effectively reduce output voltage distortion. This effectively mitigates the dependency of dead-time elimination methods on the accuracy of current polarity judgments.

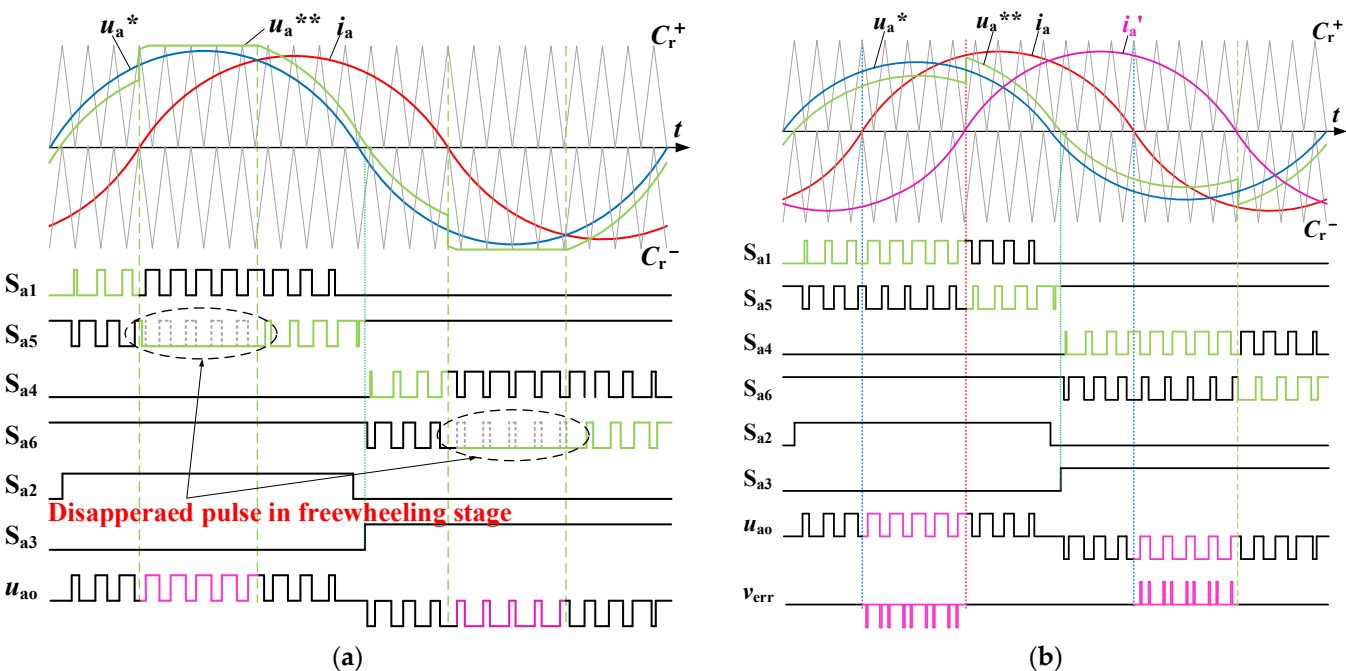

**Figure 6.** Analysis of ANPC double-modulated-wave PWM strategy. (**a**) This strategy does not introduce output voltage error when overmodulated. (**b**) This strategy can effectively reduce the output voltage error when there is a delay in current sampling.

### 3.3. Principle and Optimization Control of Midpoint Potential Balance for ANPC Three-Level PCS

First, the relationship between midpoint potential fluctuation and midpoint current in three-level ANPC topology is analyzed, as shown in Figure 7. In the figure, $U_{\text{dc1}}$ and $i_{\text{c1}}$ represent the voltage and current of the upper DC bus capacitor, respectively, while $U_{\text{dc2}}$ and $i_{\text{c2}}$ represent the voltage and current of the lower DC bus capacitor, respectively. $U_{\text{dc}}$ stands for the total voltage of the DC bus.

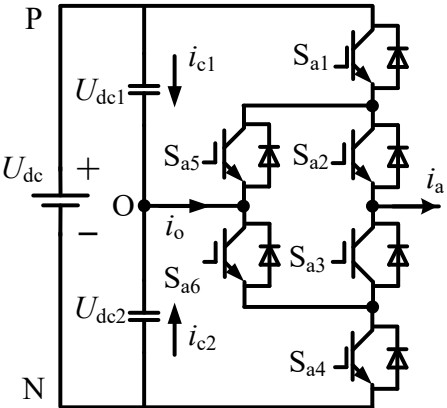

**Figure 7.** Relation between midpoint potential fluctuation and midpoint current in ANPC topology.

Assuming that the average voltage deviation between the upper and lower capacitors of the DC bus is $\Delta u_{\text{dc}}$, then we have:

$$
\begin{aligned}
i_{C1} &= C\frac{U_{\text{dc1}} - U_{\text{dc2}}}{T_{\text{s}}} = C\frac{\Delta u_{\text{dc}}}{T_{\text{s}}} \\
i_{C2} &= -C\frac{\Delta u_{\text{dc}}}{T_{\text{s}}}
\end{aligned}
\tag{5}
$$

The relationship between $\Delta u_{\text{dc}}$ and the neutral point current $i_O$ is:

$$i_O = i_{C1} - i_{C2} = 2C\frac{\Delta u_{\text{dc}}}{T_s} \tag{6}$$

The relationship between the neutral point potential and the compensating neutral point current $i_{OX}$ can be derived as:

$$i_{\text{ox}} = -i_O = -2C\frac{\Delta u_{\text{dc}}}{T_s} \tag{7}$$

Therefore, the balance control of the neutral point potential in a three-level system can be converted into neutral point current balance control. The average value of the three-phase neutral point currents within a switching period can be expressed as:

$$i_O = i_a d_{\text{ao}} + i_b d_{\text{bo}} + i_c d_{\text{co}} \tag{8}$$

$$d_{\text{io}} = 1 - |d_i|, \, (i = a, b, c) \tag{9}$$

In a three-phase three-wire symmetrical system, injecting zero-sequence voltage can alter the three phase voltages without affecting the line voltage output. The modified phase voltage is:

$$\begin{cases} u_a^* = u_a + u_z \\ u_b^* = u_b + u_z \\ u_c^* = u_c + u_z \end{cases} \tag{10}$$

By adjusting the zero-sequence compensation voltage $u_z$, $u_i$ can be modulated, subsequently influencing $d_{\text{io}}$ and ultimately regulating the neutral point current $i_O$. Consequently, by manipulating the zero-sequence voltage, we can effectively alter the neutral point current, fulfilling the objective of maintaining balanced neutral point voltage control.

The control strategy, as shown in Figure 8, involves sending the voltage difference between the upper and lower capacitors to a closed-loop PI regulator to obtain the magnitude of the zero-sequence component. Depending on the direction of the d-axis component of the three-phase current $i_d$, the zero-sequence component is added to the modulation signals of each phase, resulting in modulation waves with zero-sequence compensation.

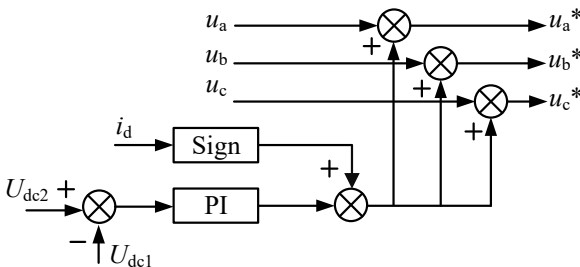

**Figure 8.** Control block diagram of neutral point potential balance.

*3.4. Narrow-Pulse Suppression Principle and Optimal Control of ANPC Three-Level PCS*

The minimum pulse width represents the smallest duration of a pulse within a fundamental wave period of the PWM pulse sequence. When the modulating pulse falls below a specified limit, known as the narrow pulse limit $t_{\text{narrow}}$, it results in a narrow pulse issue. If the generated drive width is less than $t_{\text{narrow}}$, the switching devices will fail to operate properly. The narrow pulse width can be converted to a voltage, represented as $u_{\text{narrow}}$, using the equations:

$$\begin{cases} u_{\text{narrow}} = \dfrac{U_{\text{dc}}}{2}\dfrac{t_{\text{narrow}}}{T_s} \\ u_{\text{min}} = u_{\text{narrow}} \\ u_{\text{max}} = \dfrac{U_{\text{dc}}}{2} - u_{\text{narrow}} \end{cases} \tag{11}$$

In Equation (11), $u_{\min}$ and $u_{\max}$ represent the minimum and maximum modulation voltages, respectively, that do not cause narrow pulses.

The narrow pulse regions in a three-level system are divided into two areas: when the modulation wave is close to zero, and near its amplitude. Narrow pulses are inevitable when the modulation wave is in the vicinity of zero regardless of the modulation level. When the modulation level is low, most of the modulation wave exists in this region, resulting in significant narrow pulse effects. The two narrow pulse regions near the amplitude occur only when the modulation wave is high and enters this zone. The longer the modulation wave stays in this region, the more pronounced the impact.

This paper optimizes the zero-sequence injection method based on Sine Wave Pulse Width Modulation (SPWM). The midpoint voltage $\Delta u_{\mathrm{NP}}$ deviation caused by the midpoint current is:

$$\Delta u_{\mathrm{NP}} = \frac{1}{2C} \int_{\tau}^{\tau+T_s} I_{\mathrm{NP}} \mathrm{d}t \tag{12}$$

Assuming an initial midpoint voltage deviation of $\Delta u_{\mathrm{NP0}}$, the adjusted condition should satisfy

$$\Delta u_{\mathrm{NP}} + \Delta u_{\mathrm{NP0}} = 0 \tag{13}$$

Zero-sequence voltage $U_z$ should be added such that $U_z$ is between the three-phase modulated waves to ensure $U_{\min} < U_z < U_{\max}$, corresponding to zero-sequence duty cycle $d_z$. Therefore, the adjusted three-phase modulated wave duty cycle is

$$\begin{cases} d_{\mathrm{ao}}^* = d_{\mathrm{a}} + d_z \\ d_{\mathrm{bo}}^* = d_{\mathrm{b}} + d_z \\ d_{\mathrm{co}}^* = d_{\mathrm{c}} + d_z \end{cases} \tag{14}$$

Based on the carrier comparison logic of DSP regular sampling, this paper compensates for the minimum turn-on and turn-off pulse widths during the rising and falling processes, respectively.

This paper adopts a complete narrow pulse compensation control strategy, as illustrated in Figure 9.

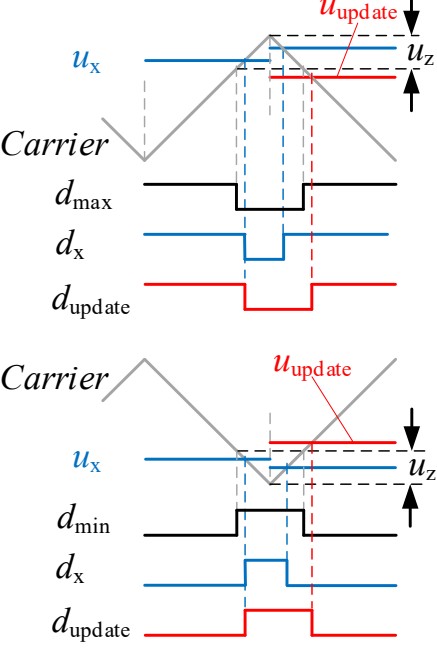

**Figure 9.** Schematic diagram of narrow pulse suppression algorithm.

In the DSP programming, modulation is implemented using the regular sampling method, with the modulation wave $u_x$ loaded at both the zero point and the periodic value of the carrier counter. When the counter is decrementing and the modulation wave is less than the minimum pulse width, the modulation wave is compensated to $u_{update}$ and used for comparison during the incrementing process. Similarly, when the counter is incrementing and the difference between the modulation wave and the maximum count value is less than the minimum pulse width, the modulation wave is also compensated to $u_{update}$ and used for comparison during the decrementing process. The compensation method is as follows:

$$u_{update} = u_i + U_z, (i = a, b, c)$$

$$U_z = \begin{cases} sign(u_i)u_{narrow} - u_i & , & |u_i| < u_{narrow} \\ 0 & , & u_{narrow} < |u_i| < \frac{U_{dc}}{2} - u_{narrow} \\ sign(u_x)(\frac{U_{dc}}{2} - u_{narrow}) - u_i & , & \left|\frac{U_{dc}}{2} - u_i\right| < u_{narrow} \end{cases}$$

$$sign(u_x) = \begin{cases} 1 & , u_i > 0 \\ -1 & , u_i < 0 \end{cases}$$

(15)

### 3.5. ANPC Three-Level PCS Nonlinear Factors Comprehensive Compensation Control

It is worth noting that in the aforementioned optimization algorithms, both the narrow pulse suppression and the neutral-point potential balancing control methods require the injection of a zero-sequence voltage. This results in an interaction between the two methods: narrow pulse suppression and neutral-point voltage balancing control both utilize zero-sequence voltage injection, leading to interactions such as the neutral-point voltage control potentially causing narrow pulse issues in other phases, and narrow pulse suppression potentially causing similar issues. Additionally, the dead-time elimination method proposed in this paper may also introduce narrow pulse problems. Therefore, a comprehensive evaluation of the effects of neutral-point potential balancing control, dead-time elimination, and narrow pulse suppression is necessary. To address this issue, this paper proposes a comprehensive compensation strategy that takes into account neutral-point potential balancing, dead-time elimination, and narrow pulse suppression. This control strategy is illustrated in Figure 10.

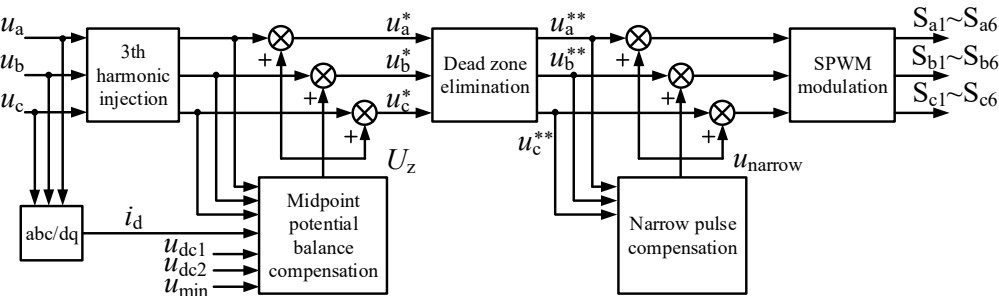

**Figure 10.** The comprehensive compensation strategy taking into account the balance of midpoint potential, dead zone elimination and narrow pulse suppression.

The design concept of this method is to restrict the regions for zero-sequence voltage injection in order to eliminate narrow pulses and dead times. By doing so, a modulation wave range that completely avoids narrow pulses and the range of injectable zero-sequence voltages can be calculated. Subsequently, in the neutral-point potential balancing control algorithm, an appropriate zero-sequence voltage component is selected and injected within the aforementioned range of zero-sequence voltages. This ensures the absence of narrow pulses and superior neutral-point potential balancing control capability.

To guarantee that the modulation wave after neutral-point potential balancing control does not generate narrow pulses, the three-phase modulation waves after adding the zero-

sequence component for neutral-point balancing control should simultaneously satisfy the following constraint conditions:

$$-1 + u_{\min} - \min(u_a, u_b, u_c) < u_z < 1 - u_{\min} - \min(u_a, u_b, u_c)$$
$$u_{\min} = u_{dt} + u_{narrow} \tag{16}$$

After using this constraint condition, the zero-sequence component limit region used by the midpoint balance control is shown in Figure 11. The overshoot headspace between the three-phase modulated wave $u_x$ and the carrier amplitude 1.0 is the overmodulation margin space. When narrow pulse compensation and dead-time elimination are not taken into account, the range of adding zero-sequence components voltage $u_Z$ for neutral point balance control corresponds to the shaded area in Figure 11a. However, considering the impact of narrow pulses, subtracting the $u_Z$ used for narrow pulse compensation from Figure 11a further results in Figure 11b, which depicts the range of zero-sequence components for neutral point balance control. It can be observed that the compensation area is divided into discontinuous regions, indicating a balance control method that compromises the neutral point balance control capability.

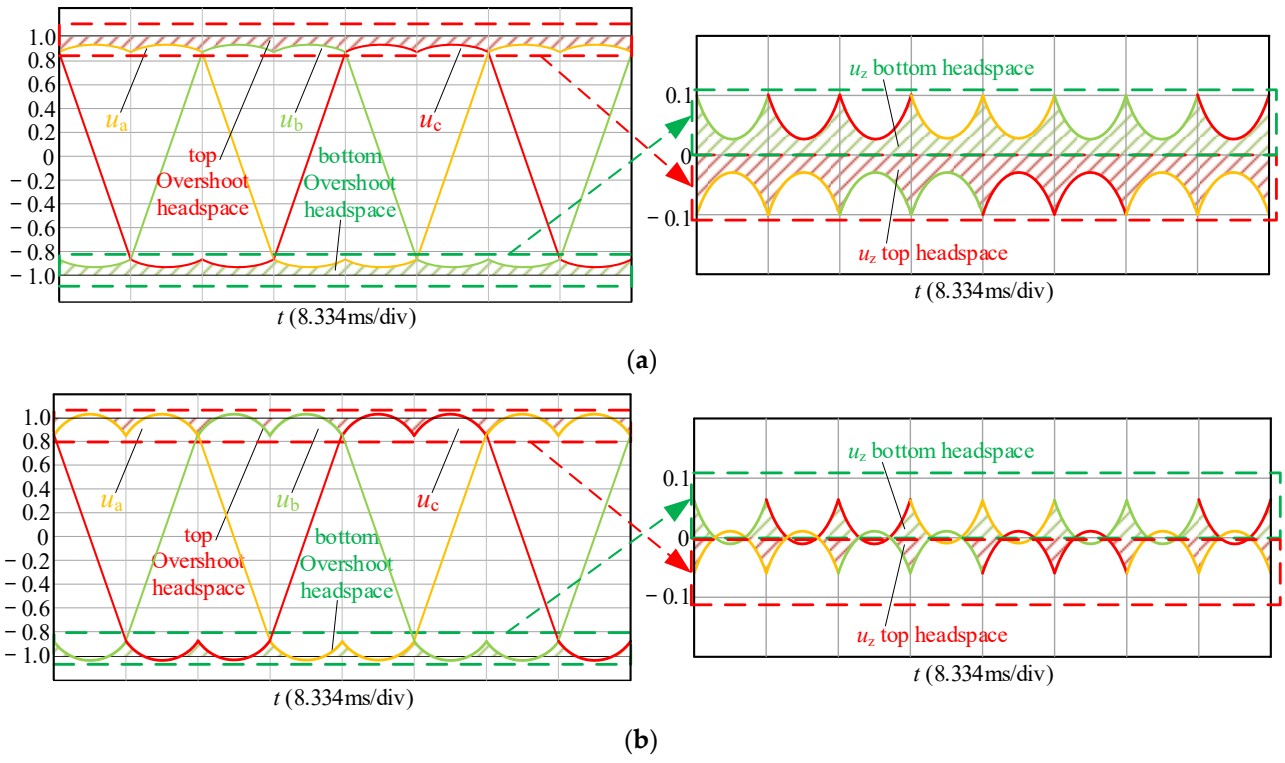

**Figure 11.** The midpoint balance controls injection into the zero-sequence component restricted region. (**a**) Zero sequence voltage injection headspace without considering narrow pulse and dead zone elimination. (**b**) Zero sequence voltage injection headspace considering narrow pulse and dead zone elimination.

## 4. Implementation and Verification of Control Strategy

As shown in Figure 12, the semi-physical simulation experimental platform used a MT6016 from a Chinese company ModelingTech (Shanghai, China) [16] as the rapid control prototype (RCP) model controller, and another MT6016 as the hardware-in-the-loop (HIL) simulator for the three-level hardware circuit. A hardware connection was established between the two devices for voltage sampling, PWM drive signal transmission, and other necessary functions.

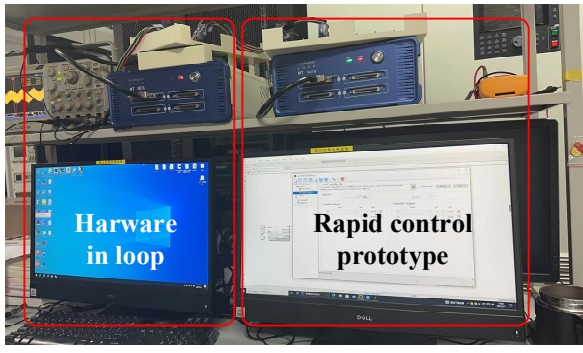

**Figure 12.** Semi-physical simulation platform.

This article takes a PCS with a rated power of 1725 kW as the simulation object, and its main electrical parameters are shown in Table 1.

**Table 1.** Electrical parameters of PCS with rated power of 1.725 MW.

| Electrical Parameter | |
| --- | --- |
| DC Voltage: 1000~1500 V | Switching Frequency: 3600 Hz |
| Grid Voltage: 690 V | Dead Time: 5 us |
| AC current: 1443 A | *L* Filter: 0.1 mH |

### 4.1. Experiment 1: Verification of Dead-Time Elimination Function

With a DC voltage set at 1200 V and a modulation index of approximately 0.94, a comparison of grid-connected currents with and without dead-time elimination, taking into account the dead-time effect, is shown in Figure 13. The experiment employed closed-loop current control to sample and synchronize the grid connection with a unit power factor, resulting in a grid-connected current of approximately 1443 A. As can be seen from Figure 13a, the dead-time effect significantly impacted the positive zero-crossing points of the current, creating small zero-voltage steps, and introducing noticeable distortion near the peak of the current. However, as shown in Figure 13b, after incorporating dead-time elimination, the current waveform appeared significantly smoother and exhibited a higher sinusoidal. A comparison of the harmonic content in Table 2 reveals that the grid-connected current harmonic distortion decreased from 2.84% to 2.15% after compensation. This comparison shows that the dead-zone elimination method can effectively reduce the 3rd–19th harmonics of the grid-connected current and improve the power quality.

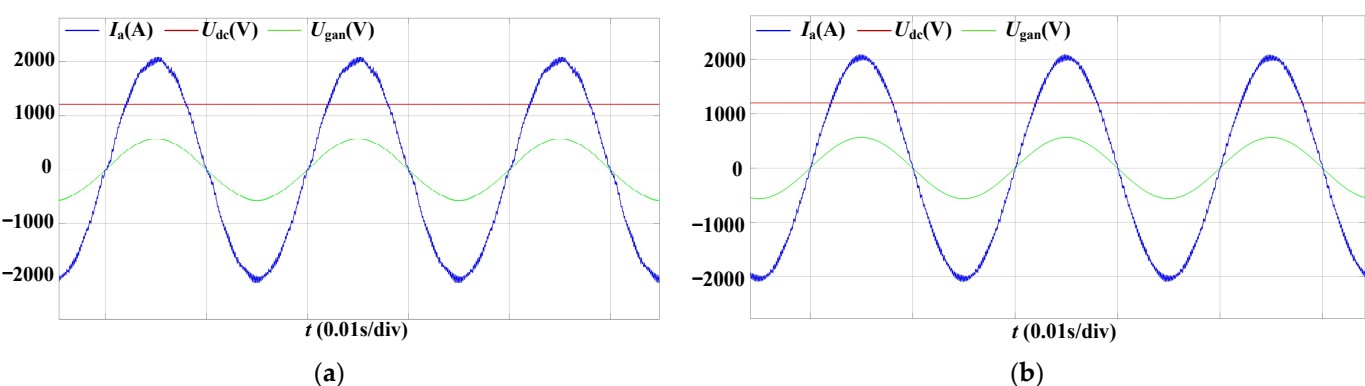

**Figure 13.** Comparison of grid-connected current and voltage waveforms, considering both the presence of dead-time effects and the application of the dead-time elimination method. (**a**) Grid-connected current and voltage waveform with dead zone effect. (**b**) Grid-connected current and voltage waveform with dead-time elimination method.

**Table 2.** Comparison of dead-time effect and dead-time elimination in grid-connected current THD.

| Harmonic Order | Dead-Time Effect % | Dead-Time Elimination % |
|---|---|---|
| THD | 2.84 | 2.15 |
| 3 | 0.2 | 0.04 |
| 5 | 0.8 | 0.12 |
| 7 | 0.6 | 0.09 |
| 9 | 0.5 | 0.01 |
| 11 | 1.21 | 0.17 |
| 13 | 0.6 | 0.1 |
| 15 | 0.16 | 0.04 |
| 17 | 0.25 | 0.04 |
| 19 | 0.15 | 0.02 |

### 4.2. Experiment 2: Functional Verification of Dead-Time Elimination under Over-Modulation Conditions

With a DC bus voltage set at 980 V and a modulation index of approximately 1.15, a comparison of grid-connected currents with and without dead-time elimination, considering the dead-time effect, is shown in Figure 14. The experiment continued to employ closed-loop current control to sample and synchronize the grid connection with a unit power factor. As shown in Figure 14a, when the dead-time effect was present, the output voltage was limited, resulting in a grid-connected current of approximately 960 A, which failed to reach the rated current and exhibited more severe current distortion. However, after incorporating dead-time elimination, the current could reach the rated value with reduced current distortion, as shown in Figure 14b. A comparison of the harmonic content in Table 3 reveals that the Total Harmonic Distortion (THD) of the grid-connected current decreased from 6.32% to 2.51% after compensation, with a significant reduction in the low-order harmonic content from the 3rd to the 19th harmonics.

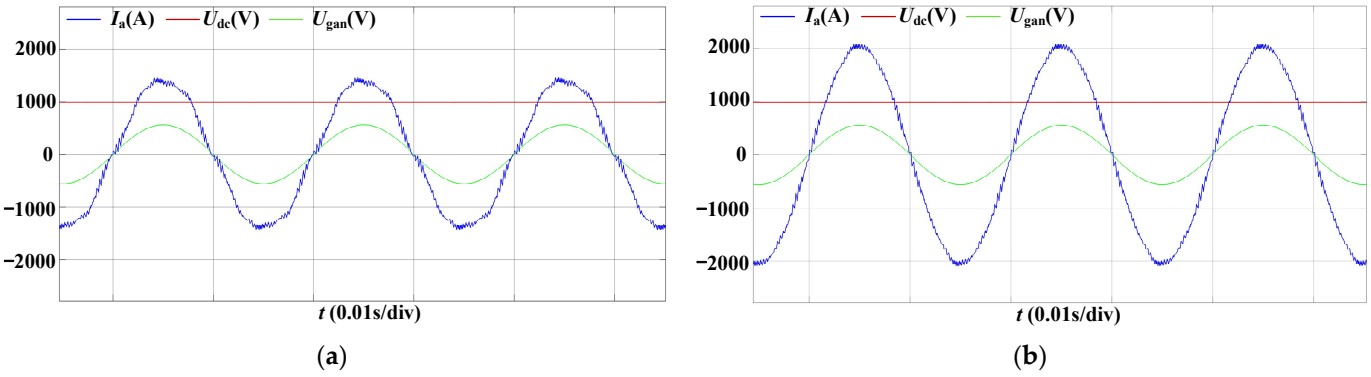

**(a)**          **(b)**

**Figure 14.** Functional verification of optimized dead-zone elimination method under an over-modulation condition. (**a**) Grid-connected current and voltage waveform with dead zone effect. (**b**) Grid-connected current and voltage waveform with dead-time elimination method.

### 4.3. Experiment 3: Functional Verification of the Improved Dead-Time Elimination Method under Current Sampling Delay

DC bus voltage set at 1200 V and current sampling delay of two computational cycles at 3.6 kHz, equivalent to 54 us, were used to simulate the delay caused by sampling calculations, filtering computations, and other factors in practical systems. A comparative experiment was conducted between the conventional dead-time elimination method and the improved dead-time elimination method under the condition of current sampling delay, as shown in Figure 15. Figure 15a demonstrates that the conventional dead-time elimination method is significantly affected by current sampling delay, rendering it unsuitable for practical use. The optimized dead-time elimination method exhibits significant

improvement in resilience to current sampling delay, validating the effectiveness of this approach, as shown in Figure 15b.

**Table 3.** THD comparison of dead-zone elimination methods before and after optimization under over-modulation conditions.

| Harmonic Order | Dead-Time Effect % | Dead-Time Elimination % |
| --- | --- | --- |
| THD | 6.32 | 2.51 |
| 1 | 100 | 100 |
| 3 | 0.4 | 0.04 |
| 5 | 4.49 | 0.69 |
| 7 | 2.18 | 0.35 |
| 9 | 0.11 | 0.04 |
| 11 | 0.53 | 0.64 |
| 13 | 1.33 | 0.27 |
| 15 | 0.09 | 0.04 |
| 17 | 0.18 | 0.08 |
| 19 | 0.1 | 0.01 |

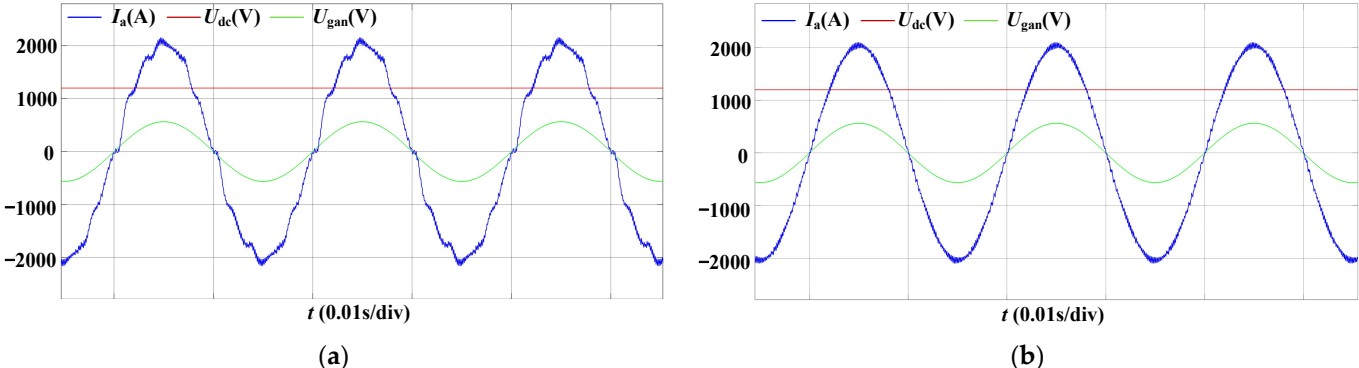

**(a)**     **(b)**

**Figure 15.** Comparison of grid-connected current before and after the optimized dead-zone elimination method is adopted under current sampling delay. (**a**) Grid-connected current and voltage waveform with conventional dead-zone elimination method. (**b**) Grid-connected current and voltage waveform with optimized dead-zone elimination method.

### 4.4. Experiment 4: Comprehensive Compensation

The experiment was conducted under the conditions of a 5 us dead time, a current sampling delay of 54 us, a DC bus voltage of 980 V, and a 10 V voltage difference added to the midpoint voltage of the DC bus. These conditions pose the risk of over-modulation and issues related to narrow pulses and unbalanced DC midpoint potential. The experiment compared and analyzed the impact of dead time, narrow pulses, and midpoint potential imbalance on the output current using an open-loop experimental approach. A comparison between the conventional SPWM modulation method and the proposed comprehensive compensation modulation method is shown in Figure 16.

Figure 16a demonstrates the optimization ability of the comprehensive compensation function for the balance of the midpoint potential of the DC bus voltage. The original 10 V voltage difference was eliminated after the compensation control is applied. However, it can be seen from the figure that there was still a slight oscillation in the DC voltage, and further research is needed on this voltage oscillation issue. Figure 16b presents the grid-connected current waveform under open-loop grid-connection conditions with the integrated compensation control proposed in this paper. Before the integration of compensation, due to the modulation index reaching 1.15, the system was at the critical state of over-modulation. Prior to compensation, the output voltage was affected by the dead zone, leading to changes in the grid-connection vector relationship. This resulted in a decrease in the amplitude of the grid-connected current and the generation of a certain amount

of reactive current. However, after incorporating integrated compensation, the impact of the dead zone on the inverter output voltage was significantly reduced, enabling the calculation of the active current based on open-loop computation. Additionally, there was a notable improvement in the Total Harmonic Distortion (THD) of the current after the integrated compensation. This demonstrates the effectiveness of the proposed comprehensive compensation strategy.

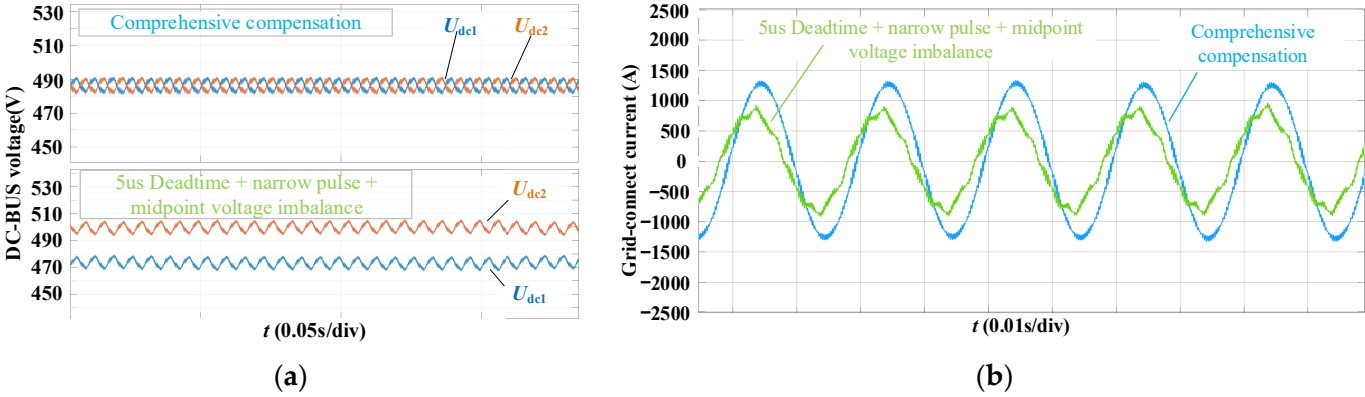

(**a**)                                                    (**b**)

**Figure 16.** Hardware-in-the-loop simulation open-loop waveform of the proposed comprehensive compensation control strategy. (**a**) The DC bus voltage waveform before and after comprehensive optimization control was adopted. (**b**) The grid-connected current waveform before and after comprehensive optimization control was adopted.

## 5. Conclusions

This article introduces an SPWM method for dead-time elimination in ANPC three-level topology. To address the issues of midpoint potential imbalance and narrow pulses under a high modulation index in three-level topologies, a comprehensive compensation control strategy is further proposed, encompassing dead-time elimination, midpoint potential balancing, and narrow pulse suppression. The proposed method enhances the modulation wave magnitude based on dead-time elimination, which in turn affects the zero-sequence component required for midpoint potential balancing and narrow pulse suppression. Prioritizing dead-time elimination, the method calculates the upper and lower limits of the zero-sequence component available for midpoint potential balancing, while ensuring complete compensation for narrow pulses. By prioritizing dead-time elimination, followed by narrow pulse suppression, and finally midpoint potential balancing, the coupling between these three factors is decoupled. The effectiveness of the proposed method was verified through hardware-in-the-loop simulations.

It should be noted that the optimization algorithm proposed in this paper still has some limitations, and focuses on the optimization of the ANPC topological dead zone elimination method. The control of the neutral point potential balance under this method still needs to be improved, and the DC voltage in this method still has a large ripple voltage, which needs to be further optimized and perfected. In addition, this method mainly adopts the Phase Disposition PWM (PD-PWM) modulation method, which could be further optimized in the future to improve the conversion efficiency of the inverter.

**Author Contributions:** Conceptualization, J.L.; methodology, J.L.; software, J.L.; validation, J.L. and J.Z.; formal analysis, J.Z.; investigation, J.Z.; resources, J.Z.; data curation, J.L.; writing—original draft preparation, J.L.; writing—review and editing, J.L.; visualization, J.L.; supervision, J.Z. project administration, J.L.; funding acquisition, J.Z. All authors have read and agreed to the published version of the manuscript.

**Funding:** This research was funded by the National Key R&D Program of China (2021YFE0103800).

**Data Availability Statement:** Data is contained within the article.

**Conflicts of Interest:** The funders had no role in the design of the study; in the collection, analyses, or interpretation of data; in the writing of the manuscript; or in the decision to publish the results.

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
