# Peer review of "Waveform Optimization Control of an Active Neutral Point Clamped Three-Level Power Converter System"

_electronics, doi:10.3390/electronics13101980_

Round 1

Reviewer 1 Report

Comments and Suggestions for Authors

This paper presents a compensation control strategy combining dead-time elimination, midpoint potential balance, and narrow pulse suppression based on the active neutral point clamped (ANPC) three-level topology. Semi-physical simulations are conducted to verify the proposed method. The study is of certain significance. However, it needs major revision before acceptance. 

1. The contributions of the paper should be made clear, in the introduction.

2. The variables in equations should be explained. Please check throughout the paper.

3. Four plots are given in Figure 3. They should be explained in detail.

4. The format of Table 1 should be revised.

5. It is difficult to understand Figure 16 since nothing can be read from it.

6. The limitaion of the study should be emphasized. 

Comments on the Quality of English Language

Minor editing of English language is required.

Reviewer 2 Report

Comments and Suggestions for Authors

Authors report a control strategy applied to a power converter system. They combine dead-time elimination, midpoint potential balance, and narrow pulse suppression based on a neutral point clamped three-level topology. Simulation results are presented to validate their proposed methodology.

In my opinion, the general structure of the work is adequate, however, I also consider that it lacks of mathematical support and references; i. e., apart from the introduction section, no reference is included for development their work. In addition, no section title highlights the main contributions.

How did the authors compare their results to the state of the art? How did they conclude that their results are better or improve than the results of the other published papers? What are those studies that they compared their results to?

Round 2

Reviewer 1 Report

Comments and Suggestions for Authors

My concerns have been well addressed. I recommend it to be accepted.

Comments on the Quality of English Language

 Minor editing of English language is required.